



# Statistical Hypothesis Testing in Wavelet Analysis: Theoretical Developments and Applications to India Rainfall

Justin A. Schulte[1]

[1]Science Systems and Applications, Inc,, Lanham, 20782, United States

*Correspondence to*: Justin A. Schulte (justin.a.schulte@nasa.gov)

**Abstract** Statistical hypothesis tests in wavelet analysis are reviewed and developed. The output of a recently developed cumulative area-wise is shown to be the ensemble mean of individual estimates of statistical significance calculated from a geometric test assessing statistical significance based on the area of contiguous regions (i.e. patches)

of point-wise significance. This new interpretation is then used to construct a simplified version of the cumulative area-wise test to improve computational efficiency. Ideal examples are used to show that the geometric and cumulative area-wise tests are unable to differentiate features arising from singularity-like structures from those associated with periodicities. A cumulative arc-wise test is therefore developed to test for periodicities in a strict sense. A previously proposed topological significance test is formalized using persistent homology profiles (PHPs) measuring the number

of patches and holes corresponding to the set of all point-wise significance values. Ideal examples show that the PHPs can be used to distinguish time series containing signal components from those that are purely noise. To demonstrate the practical uses of the existing and newly developed statistical methodologies, a first comprehensive wavelet analysis of India rainfall is also provided. A R-software package has been written by the author to implement the various testing procedures.

**1. Introduction**

Time series describing the evolution of physical quantities such as streamflow, sea surface temperature (SST), rainfall, and wind speed often contain non-stationary and time-scale dependent characteristics. A better understanding of these characteristics is facilitated through the application of various statistical and signal processing methods that account for them. One such method is wavelet analysis, which is a time-frequency analysis method for

extracting time-localized and scale-dependent features from time series. This method contrasts with the widely known Fourier analysis that assumes stationarity. The short time Fourier transform (STFT) addresses the problem of time-localization, but wavelet analysis is still preferred over the STFT because wavelet analysis uses a variable window width that more effectively separates signal components. An additional attractive aspect of wavelet analysis is that it can be used to quantify the relationship or coherence between two time series at an array of time scales in a non-

stationary setting (Grinsted et al., 2004). More recently, the frequency domain analogs of partial and multiple correlation (Ng and Cha, 2012; Hu and Si, 2016) have been developed in wavelet analysis, making the method an even more powerful exploratory tool for researchers. Given these desirable  aspects of wavelet analysis, it is not surprising that wavelet analysis has been applied in a broad range of topics, including climatology (Gallegati, 2018), hydrology (Schaefli et al., 2007; Labat, 2010), forecast model verification (Lane, 2007; Liu et al., 2011), ensemble

forecasting (Schulte and Georgas, 2018), and biomedicine (Addison, 2005).



One application of wavelet analysis is the estimation of a sample wavelet spectrum and the subsequent comparison of the sample wavelet spectrum to a background noise spectrum. To make such comparisons, one must implement statistical tests. Torrence and Compo (1998) were the first to place wavelet analysis in a statistical hypothesis testing framework using point-wise significance testing. In the point-wise approach, the statistical

significance of wavelet quantities associated with points in a wavelet spectrum are assessed individually without considering the correlation structure among wavelet coefficients. For wavelet power spectra of climate time series, theoretical red-noise spectra are often the noise background spectra against which sample wavelet power spectra are tested. Monte Carlo methods are used to estimate the background noise spectra for wavelet coherence (Grinsted et al. 2004), partial coherence (Ng and Cha, 2012), multiple coherence (Hu and Si, 2016), and auto-bicoherence (Schulte,

2016b).

Despite its wide use, the point-wise approach has two drawbacks, the first of which is that the test will frequently generate many false positive results because of the simultaneous testing of multiple hypotheses. The second drawback is that spurious results occur in clusters because wavelet coefficients are correlated. To account for these deficiencies, Maraun et al. (2007) developed an area-wise test to reduce the number of spurious results. Additional

tests were subsequently developed by Schulte et al. (2015) and Schulte (2016a) to address the deficiencies of the point-wise test and computational inefficiencies of the area-wise test. Although these tests were demonstrated to be effective at reducing spurious results, the point-wise testing procedure is still more frequently adopted. Furthermore, there are numerous papers surveying general aspects of wavelet analysis (Meyers et al., 1993; Kumar and Foufoula-Georgiou, 1997; Torrence and Compo, 1998; Labat, 2005, Lau and Weng, 1995; Addison, 2005; Schaefli et al., 2007; Sang et

al., 2012), but no papers surveying the recent developments in statistical hypothesis testing. This observation underscores the need for a paper that summarizes theoretical and practical aspects of statistical hypothesis testing.

A physical application to which wavelet methods have been applied is the understanding of India rainfall variability. India rainfall variability is a complex, non-stationary, and time-scale dependent phenomenon, making wavelet analysis a well-suited tool for studying it. Recognizing the non-stationary behavior of the Indian monsoon

phenomena, Torrence and Webster (1999) used wavelet coherence analysis to show that the relationship between the El Nino/Southern Oscillation (ENSO) and Indian rainfall is strong and non-stationary in the 2 to 7-year period band. Narasimha and Bhattacharyya (2010) used wavelet cross-spectral analysis to link the solar cycle to changes in the India monsoon. Other studies have used wavelet analysis to understand the temporal characteristics of the India rainfall time series (Nayagam et al. 2009). Fasullo (2004) found biennial oscillations in the all-India rainfall time series, while

Yadava and Ramesh (2007) found significant long-term periodicities in an India rainfall proxy time series. Terray et al. (2003) found that a time series describing late summer (September-August) India rainfall is associated with significant wavelet power in the 2 to 3 year period band and suggested such power is related to the tropospheric biennial oscillation. Common to all the studies noted above is the use of point-wise significance testing. Recent work highlighting the pitfalls of the point-wise testing approach raises the question as to whether the features identified in

previous wavelet studies of India rainfall are statistical artifacts or ones distinguishable from the background noise.





To address this question, an additional study is needed that applies the new statistical hypothesis tests in wavelet analysis.

In this paper, a first survey of the theoretical and practical aspects of recent advances in statistical significance testing of wavelet estimators is presented in Section 2. Section 2 also includes discussions about the modifications of existing tests designed to make them more computationally efficient than the original formulations. A cumulative arc-wise test is also proposed for testing for the presence of periodicities embedded in noise in a strict sense. Section 3 is devoted to the presentation of a comprehensive wavelet analysis of India rainfall time series using recently developed wavelet methods. The paper concludes with a summary and discussion in Section 4.

## 2. Wavelet Analysis

### 2.1 Basic Overview

The continuous wavelet transform of a time series $X(t)$ is given by

$$W(b,a) = \frac{1}{\sqrt{a}} \int_{-\infty}^{\infty} X(t) \psi^* \left( \frac{t-b}{a} \right) dt, \tag{1}$$

where $a$ is wavelet scale, $\psi$ is an analyzing wavelet, and $b$ is time. The sample wavelet power spectrum is $|W(b,a)|^2$ and measures the energy content of a signal at time $b$ and scale $a$. Thus, the wavelet transform of a time series produces a two-dimensional representation of it. In this paper, the set consisting of all points in the two-dimensional representation will be denoted by $\mathbb{H}$ and referred to as the time-scale plane. To simplify the discussion of results in this paper, the commonly used Morlet wavelet with angular frequency $\omega = 6$ is used throughout. For more details about wavelet analysis, the reader is referred to Torrence and Compo (1998).

Unlike the Fourier analysis where neighboring frequencies are uncorrelated, the wavelet coefficients at neighboring points in $\mathbb{H}$ are intrinsically correlated. The intrinsic correlation between wavelet coefficients at $(b, a)$ and $(b', a')$ is represented by the reproducing kernel $K(b, a, b', a')$ whose mathematical expression is

$$K(b,a;b',a') = \frac{1}{c_\psi \sqrt{aa'}} \int \left[ \psi \left( \frac{t-b'}{a'} \right) \psi^* \left( \frac{t-b}{a} \right) \right] dt, \tag{2}$$

where $c_\psi$ is an admissibility constant. The redundancy between the values $W(a,b)$ and $W(a',b')$ is expressed as

$$W(b,a) = \iint K(b,a;b',a') W(a',b') \frac{da'}{a'^2} db'. \tag{3}$$

Eq. (3) says that a wavelet coefficient at $W(b,a)$ captures information from neighboring points, the degree to which depends on the weight $K(b,a;b',a')$. Even for uncorrelated noise, wavelet coefficients will be correlated in $\mathbb{H}$ (Maraun and Kurths, 2004), a theoretical result that has important implications for significance testing.

The normalized reproducing kernel for the Morlet wavelet is shown in Figure 1. In Figure 1a, the reproducing kernel is dilated and translated to the scale $a = 32$ and time $b = 500$ and indicates that a wavelet coefficient at (500,32) will be correlated with other wavelet coefficients surrounding it. The reproducing kernel shown in Figure 1b is dilated





and translated to (500, 128) and seen to be wider in the time direction than the reproducing kernel centered at (500, 32). The widening reflects how the reproducing kernel expands linearly in both the time and scale (in a non-logarithmic scale) direction.

## 2.2 Statistical Significance tests

### 2.2.1 Pointwise significance

In point-wise significance testing, one individually compares a wavelet quantity at every point in $\mathbb{H}$ to the critical level of the point-wise test, which depends on the chosen point-wise significance level $\alpha_{pw}$ and usually on the wavelet scale $a$. For wavelet power spectra of climate time series, point-wise test critical values are often determined from a theoretical red-noise background (Torrence and Comp, 1998). For wavelet coherence, partial coherence,

multiple coherence, and auto-bicoherence, Monte Carlo methods need to be implemented to estimate the critical values (Grinsted et al., 2004; Ng and Chan, 2012; Hu and Si, 2016, Schulte, 2016a). However, a parametric bootstrap method may be required for determining the critical values of an arbitrary background model if analytical background models are not readily available (Maraun et al., 2007). Using the point-wise test, one assigns to each point in $\mathbb{H}$ a p-value, $\rho_{pw}$, representing the probability of finding the observed or more extreme wavelet quantity (power, coherence, etc.)

when the null hypothesis is true. The result of the point-wise test is the subset

$$P_{pw} = \{(b, a): \rho_{pw}(b, a) < \alpha_{pw}\} \tag{4}$$

of $\mathbb{H}$ representing regions where point-wise significant wavelet quantities have been identified.

To better understand the utility of the point-wise testing procedure, consider two example time series, where the first time series , $R(t)$, corresponds to a realization of a red-noise process with lag-1 auto-correlation coefficient

equal to 0.4 (Figure 2a). The second time series, $X(t)$, shown in Figure 3a is given as

$$X(t) = S(t)/\sigma_s + N(t)/m\sigma_n, \tag{5}$$

where

$$S(t) = \sum_{j=1}^{3} \sin \frac{2\pi}{2^{j+3}} t + \delta(t), \tag{6}$$

$$\delta(t) = \begin{cases} 60 & t = 1000 \\ 0 & otherwise \end{cases}, \tag{7}$$

and $N(t)$ is a realization of a red-noise process with lag-1 auto-correlation coefficient equal to 0.1. The constants $\sigma_s$ and $\sigma_s$ are the standard deviations of $S(t)$ and $N(t)$, respectively. The real number $m$ is a measure of the signal-to-noise ratio, larger values indicating relatively more signal.

The outcomes of the point-wise test applied to the (rectified; Liu et al., 2007) wavelet power spectrum of $X(t)$ and $R(t)$ are shown in Figures 2b and 3b. For $R(t)$, the point-wise test applied at $\alpha_{pw} = 0.05$ identified many statistically

significant wavelet power coefficients, all of which are spurious by construction. The spurious results are seen to occur



in contiguous regions, and the union of all such regions is $P_{pw}$. For the wavelet power spectrum of $X(t)$, statistically significant wavelet power at the periods 64, 128, and 256 is seen to cluster in narrow bands, reflecting the periods of the individual sinusoids (Figure 3b). The singularity at $t = 1000$ emerges as a scale-elongated region of point-wise significance. All other significance regions are spurious, as those are associated with $N(t)$. Thus, without further

investigation, it would be impossible to know without prior knowledge of $X(t)$ if the features associated with $N(t)$ are part of the signal or not. These examples highlight how the number of spurious results can be large and how they could impede the differentiation between an underlying signal and background noise. It is therefore important to reduce the number of spurious results using other statistical methods.

### 2.2.2. Area-wise and Geometric Testing

As noted by Maraun et al. (2007), the application of the point-wise test $N_{pw}$ times will on average produce $N_{pw}\alpha_{pw}$ spurious results, with the spurious results occurring in clusters or patches. These so-called point-wise significance patches arise from the reproducing kernel of the analyzing wavelet that represents intrinsic correlations among wavelet coefficients. As noted by Schulte (2016), patches can be rigorously defined using ideas from topology. That is, a patch is a path-connected component of $P_{pw}$, where a path-connected component is an equivalent class of

$P_{pw}$ resulting from an equivalence relation $\sim$ on $P_{pw}$ that makes points $x, y \in P_{pw}$ equivalent (written $x \sim y$) if they can be connected by a continuous path $f : [0\ 1] \rightarrow P_{pw}$ such that $f(0) = x$ and $f(1) = y$ (Figure 4). The equivalence relation $\sim$ reduces the original large set of points in $P_{pw}$ to a smaller set of patches, the implications of which will be described later.

Because patches arise from the reproducing kernel, patches inherit the geometric characteristics of the

reproducing kernel such as convexity, area, length, and width. As a concrete example, consider the set shown in Figure 1b consisting all of points enclosed by the thick black contour. The contoured region is subset of $\mathbb{H}$ for which the normalized reproducing kernel dilated and translated to (500,128) exceeds 0.1. The set is convex (i.e. contains no concavities) because any two points in it can be connected by a line that remains entirely in the set (Figure 4). The convexity of the set suggests that typical patches will be convex, which can be confirmed by generating a large

ensemble of patches found in the wavelet power spectra of realizations of a red-noise process (Schulte et al., 2015). The convexity of patches plays an important role in understanding how different statistical tests perform. Another important geometric property is area, which reflects the dilated reproducing kernel so that patches at greater scales will be generally larger than those located at smaller scales, as Figure 1 suggests.

Recognizing that a typical patch area is given by the reproducing kernel, Maraun et al. (2007) developed an

area-wise test, which is conducted as follows: First choose a critical area $P_{crit}(b, a)$ defined as a subset of $\mathbb{H}$ for which the reproducing kernel dilated and translated to $(b, a)$ exceeds a certain critical level $K_{crit}$. That is,

$$P_{crit}(b, a) = \{(b', a')|\ K(b, a; b', a') > K_{crit}\} \tag{8}$$

The set of points whose associated wavelet quantities are also area-wise significant is then the set



$$P_{aw} = \bigcup\nolimits_{P_{crit}(b,a) \subset P_{pw}} P_{crit}(b,a) \tag{9}$$

representing the union of all critical areas that lay completely inside $P_{pw}$. The larger the critical area, the larger a patch needs to be for it deemed area-wise significant. Thus, $P_{crit}(b,a)$ is related to $\alpha_{aw}$, the significance level of the area-wise test. As discussed by Maraun et al. (2007), the critical area that corresponds to $\alpha_{aw}$ is determined using a root-finding algorithm. This step is non-trivial but can be circumvented by performing a geometric test instead, as discussed below.

While the area-wise effectively addresses the multiple-testing pitfall of the point-wise test, the use of the root-finding algorithm renders difficult the practical implementation of the test. To overcome this drawback, Schulte et al. (2015) constructed a geometric test whose test statistic is normalized area. The normalized area of a patch is defined as the patch area, $A_{patch}$, divided by the square of the patch's mean scale coordinate, $\hat{a}$. That is,

$$A_{norm} = \frac{A_{patch}}{\hat{a}^2}, \tag{10}$$

where the division by the mean scale coordinate accounts for how the reproducing kernel results in the scale-dependent expansion of patches in the time and scale direction. Thus, the normalized areas of patches can be readily compared regardless of their location in $\mathbb{H}$. The critical value of this test is assessed using Monte Carlo methods as follows: (1) Generate wavelet spectra under some null hypothesis (e.g. red noise); (2) create a null distribution of normalized area using the patches found in the wavelet spectra; and (3) estimate the critical level of the test corresponding to the geometric significance level $\alpha_{geo}$ by computing the $100(1-\alpha_{geo})$-th percentile of the null distribution. This null distribution calculation can be performed rapidly because wavelet spectra often contain many patches. However, for wavelet coherence, Monte Carlo methods must be applied twice because the critical levels of the point-wise test must be also empirically estimated. Fortunately, the length of the noise realizations needed to generate the null distribution of normalized areas need not be the same length of the input time series because patch area is unrelated to the time series length; it is related to the reproducing kernel. As such, the realizations can be of shorter length, improving the efficiency of the null distribution computation. The ability to efficiently generate a null distribution allows p-values associated with the geometric test to be further adjusted to account for multiple testing. For example, the false discovery rate of the geometric test can be controlled at a desired level if the number of patches to which the test is applied is large (Schulte et al., 2015).

It is important to note that the geometric and area-wise tests differ in the way they assign statistical significance to points. For the geometric test, all points in a patch will be deemed (in)significant if the patch is deemed statistically (in)significant. However, strictly speaking, the area-wise test evaluates the statistical significance of wavelet quantities associated with points in $P_{pw}$ based on the geometric properties of the path-connected subsets of $P_{pw}$ to which they belong. That is, for a wavelet quantity at $(b, a)$ to be area-wise significant means that there must be a $P \subseteq P_{pw}$ containing $(b, a)$ that is path-connected, sufficiently convex, and large to have $P_{crit}(b,a) \subset P$. If $P$ is a patch, then no subset of $P$ can be area-wise significant if $P$ is not, consistent with the patch-sorting interpretation. On the other hand, $P$ may contain both area-wise and non-area wise significant subsets, opposing the patch-sorting



interpretation. As discussed earlier, the equivalent relation ~ reduces the initial large set of points to a set of fewer patches, implying that the number of spurious results arising from the geometric and area-wise test is less than that of the point-wise test.

Figures 2 and 3 show the wavelet power spectra of $X(t)$ and $R(t)$ after the application of the area-wise and

geometric tests at the 0.1 significance level, where the area-wise test was performed using an existing R-software package (available at: https://rdrr.io/github/Dasonk/SOWAS/ ). Both tests are seen to reduce the number of spurious results arising from the point-wise significance test applied at the $\alpha_{pw}= 0.05$ level. However, as shown in Figure 3c, the area-wise test also deems features associated with the signal $S(t)$ as noise. For example, the patch located at a period of 256 extending from $t = 100$ to $t = 1700$ is no longer statistically significant. Note also that the area-wise test

largely filters out the scale-elongated feature associated with the singularity of $X(t)$ at $t = 1000$, while the geometric test deems the feature statistically significant. This difference in performance is expected because patches must be long with respect to the reproducing kernel to be area-wise significant, whereas there is no such constraint for the geometric test. Thus, the area-wise is better suited for situations in which only periodicities are sought because periodicities induce temporally long patches. In general, the area-wise and geometric tests will reduce the signal

detected by the point-wise test because the area-wise and geometric tests are theoretically constrained to detect no more signal than the point-wise test (Table A.1). This theoretical constraint raises a natural question: can a test simultaneously reduce statistical artifacts and increase signal detection (true positive results) relative to the point-wise test? The fact that patches enlarge with increasing $\alpha_{pw}$ suggests that examining the areas of patches across a set of point-wise significance levels could improve signal detection relative to the point-wise test.

**2.2.3. Cumulative Area-wise Testing**

Maraun et al. (2007) and Schulte et al. (2015) demonstrated that the area-wise and geometric tests are procedures for reducing the number of spurious results. However, both tests suffer from the drawback of having to select both an area-wise (or geometric) significance level and a point-wise significance level. This dual significance level selection is a cause for concern because a single wavelet quantity can have different levels of area-wise or

geometric significance depending on the chosen point-wise significance level, leading to uncertainty as to whether the wavelet quantity is statistically artificial or distinguishable from background noise.

To address this concern, Schulte (2016) suggested that changes in the geometric and topological characteristics of patches should be assessed over a finite set of point-wise significance levels $\alpha_1, \alpha_2, ..., \alpha_N$. If the point-wise significance levels $\alpha_1, \alpha_2, ..., \alpha_N$ are chosen such that

$$\alpha_1 < \alpha_2 < \cdots < \alpha_N, \tag{11}$$

then the sets

$$P_{pw}^i = \{(b,a): \rho_{pw}(b,a) < \alpha_i\}, \tag{12}$$

will form a filtration





$$\emptyset = P_{pw}^1 \subseteq P_{pw}^2 \subseteq \cdots \subseteq P_{pw}^N = \mathbb{H}, \qquad (13)$$

of $\mathbb{H}$ for sufficiently small $\alpha_1$ and large $\alpha_N$. Eq. (13) expresses the intuitive idea that one can start with an empty set (no statistically significant wavelet quantities) and build the complete time-scale plane by increasing the point-wise significance level to a value that renders all wavelet quantities statistically significant.

5        An idealized example of a wavelet domain filtration is shown in Figure 5a. The application of the point-wise test at the $\alpha_1$ level results in no statistically significance regions, whereas the application of the test at $\alpha_2$ results in a relatively small region of point-wise significance (dark gray). In contrast, setting the point-wise significance level to $\alpha_5$ results in nearly all wavelet quantities being statistically significant, as indicated by the light gray region. At $\alpha_6$, all wavelet quantities are statistically significant, and $P_{pw}^6$ is $\mathbb{H}$.

10        Although filtrations like Eq. (13) haven proven useful in a data analysis method called persistent homology (Edelsbrunner and Harer, 2008), it focuses on the sets of all point-wise significant values. At present, the concern is the statistical significance of a wavelet quantity at a point, and thus a more local approach is needed. The utility of Eq. (13) will become more apparent in Section 2.2.5. To localize the persistence approach, a local filtration about $x$ called a geometric pathway is needed (Schulte, 2016). Mathematically, a geometric pathway $G_x$ corresponding to a point $x$
15 is a nested sequence

$$\emptyset = P_1^x \subseteq P_2^x \subseteq \cdots \subseteq P_N^x = \mathbb{H}, \qquad (14)$$

where

$$P_i^x = \left\{ (b,a) : (b,a) \in P_{pw}^i, (b,a) \sim x \right\}. \qquad (15)$$

       A concrete example of a geometric pathway $G_x$ is shown in Figure 5b, where the geometric pathway
20 corresponds to the wavelet domain filtration shown in Figure 5a. In this example, $P_1^x = \emptyset$ and $P_2^x = \emptyset$, where $P_2^x = \emptyset$ because $x$ is not in the dark gray region shown in Figure 5a representing $P_{pw}^2$. Although the set $P_{pw}^3$ comprises two path-components or patches, only the component containing both $x$ and $y$ corresponds to the geometric pathway member at $\alpha_3$. The reason is because any point equivalent to $w$ is not equivalent to $x$ on $P_{pw}^3$, as those points cannot be connected to $x$ by a path fully contained in $P_{pw}^3$. Thus, in this example,

$$P_3^x = \left\{ (b,a) : (b,a) \in P_{pw}^3 \; and \; (b,a) \sim x \right\}, \qquad (16)$$

$y \in P_3^x$, and $w \notin P_3^x$. A similar situation occurs at $\alpha_4$ because $P_{pw}^4$ has three path-connected components containing $v$, $w$, $x$, $y$, and $z$, but $x$ is only equivalent to points that are equivalent to $y$. At $\alpha_5$, the set $P_{pw}^5$ is path-connected ($x \sim y \sim z \sim v \sim w$) and is thus the 5th member of $G_x$. The 6th and last member of $G_x$ is $\mathbb{H}$.

       Using the definition of a geometric pathway, Schulte (2016) constructed a cumulative area-wise test that
30 assesses the statistical significance of wavelet quantities by associating to each point $x$ in $\mathbb{H}$ the total normalized area integrated over its geometric pathway $G_x$. The first step of the procedure is to calculate the normalized areas





$A_1^x, A_2^x, \dots, A_N^x$ corresponding to the $N$ members of $G_x$, where $A_i^x$ is assumed to be zero if $P_i^x = \emptyset$ or $P_i^x = \{x\}$. The second step is to compute the sum

$$A^x = \sum_{k=1}^{N} A_k^x \tag{17}$$

and compare $A^x$ to a critical area $A_{crit}$ determined by Monte Carlo methods. In the Monte Carlo approach, wavelet

spectra and corresponding geometric pathways are generated under some null hypothesis and a null distribution is ensembled from all the computed total normalized areas. The critical area corresponding to the $\alpha_c$ level of the test is the $100(1-\alpha_c)$-th percentile of the null distribution. Performing these steps for all $x \in \mathbb{H}$ results in an adjusted $p$-value at every point that accounts for multiple testing and the correlation among adjacent wavelet coefficients. As noted by Schulte (2016), the computational efficiency of the cumulative area-wise test can be improved by setting $\alpha_1 = 0.02$

and $\alpha_N = 0.18$, though this selection means that the cumulative area-wise test may no longer adjust $p$-values at every point in $\mathbb{H}$ because complete filtrations like Eq. (13) may no longer be under consideration. Nevertheless, using a limited range of point-wise significance levels generally results in a better performance of the cumulative area-wise test relative to the geometric test performed at a single point-wise significance level (Appendix A and Table A.1).

A disadvantage of the cumulative area-wise testing procedure is that it involves the computation of nested

sequences, which renders the test computational inefficient for long time series. Fortunately, the computation of nested sequences can be avoided as follows. First note that the test statistic for the cumulative area-wise test can be replaced by mean normalized area without changing the outcome of the test. The criterion for a point to be cumulative area-wise significant is then

$$\hat{A}^x = \frac{1}{N} \sum_{k=1}^{N} A_k^x > \hat{A}_{Crit}, \tag{18}$$

where $\hat{A}_{Crit}$ is the critical level of the test calculated by computing the $100(1 - \alpha_c)$-th percentile of the null distribution comprising mean normalized areas. Furthermore, there is a one-to-one relationship between the normalized area of a patch and its geometric test p-value because a greater normalized area always means a lower p-value. Thus, if $\rho_i^x$ is the p-value associated with $A_i^x$ then the criterion for a wavelet quantity to be cumulative area-wise significant is

$$\hat{\rho}^x = \frac{1}{N} \sum_{k=1}^{N} \rho_k^x < \alpha_c \tag{19}$$

because there is also a one-to-one relationship between $\alpha_c$ and the critical level of the test. Eq. (19) implies that to perform the cumulative area-wise test, one can individually perform the geometric test at each point-wise significance level and then compute $\hat{\rho}^x$, avoiding the computation of nested sequences as in the original formulation of the cumulative area-wise test.

This interpretation of the cumulative area-wise test also allows the testing method to be put in an ensemble forecasting framework. That is, the individual geometric test p-values are identified as ensemble members estimating the statistical significance of a wavelet quantity. The mean p-value is identified with the ensemble mean that smooths out the unpredictable aspects of the forecast. In the present context, $\hat{\rho}^x$ smooths out the features in the wavelet spectra





resulting from noise and leaves the features on which most of the geometric tests agree are significant. This smoothing results in the suppression of noise but enhancement of signal. These ideas are consistent with how Schulte (2016) found the cumulative area-wise test to perform superiorly to the geometric test in terms of signal detection, at least for periodic signals (Appendix A and Table A.1).

5    Using Eq. (19) we now devise a simplified approach to the cumulative area-wise test. In the simplified approach, all points in $\mathbb{H}$ are initially assigned a value of 0. Then, for a fixed point-wise significance level, the geometric test is performed on all patches. The points falling in those patches that are geometrically significant at the $\alpha_c$ level are then assigned the value 2. The values for the points not in geometrically significant patches are left unchanged. This assignment step is then repeated for each point-wise significance level and the values are stored

10   separately so that each point in $\mathbb{H}$ is assigned a set of values, the number of values equaling the number of point-wise significance levels used to implement the cumulative area-wise test. Then, the average value at each point is computed. Regions where the average value exceeds unity are then regions of cumulative area-wise significance.

   The application of the simplified cumulative area-wise test to the wavelet power spectra of $R(t)$ and $X(t)$ (Figures 2e and 3e) reveals that the test is effective at reducing the number of spurious results arising from the point-

15   wise test when applied at the $\alpha_c = 0.05$ level. Like the area-wise test, the reduction in spurious features is accompanied by a loss in signal detection. For example, the cumulative area-wise test, like the area-wise test, deems the wavelet power associated with the point-wise patch located at a period of 256 extending from $t = 100$ to $t = 1700$ in Figure 3b as statistically insignificant. On the other hand, the scale-elongated feature associated with the singularity of $X(t)$ at $t = 1000$ more clearly emerges from the application of the cumulative area-wise test than the area-wise test, which is

20   not surprising because the cumulative area-wise test assesses statistical significance based on area without regard to the specific shape of patches.

   Although this example suggests the point-wise detects more signal than the cumulative area-wise test, a simplified experiment in Appendix A revealed that the simplified cumulative area-wise test can detect more signal than the point-wise test when the signal-to-noise ratio is relatively high (Table A.1). At low point-wise significance

25   levels, the tests were found to detect a similar amount of signal on average. Thus, a key strength of the cumulative area-wise test is that it can retain signal, while also reducing the number of spurious results arising from the point-wise test.

### 2.2.4. Cumulative Arc-wise Testing

   While the area-wise, geometric, and cumulative area-wise tests can effectively reduce the number of spurious

30   results, they do not test for the existence of periodicities embedded in a time series. This deficiency is especially true for the geometric and cumulative area-wise tests because they only consider the area of patches in the assessment of statistical significance. For example, the cumulative area-wise test can identify features that are long in scale but short in time as statistically significant (Figure 3e). Thus, a new test needs to be constructed that is specifically designed to distinguish randomly stable fluctuations from those that are periodicities. As periodicities are determined by patch

35   length, a strict test for periodicities should consider patch length and not patch area.



Using the idea that patch length is more closely linked to the presence of periodicities, we construct a cumulative arc-wise test that can be implemented using the following steps. For Step 1, fix a scale and point-wise significance level and identify point-wise significance arcs, where point-wise significance arcs are contiguous strings of points whose associated wavelet quantities are point-wise significant. Step 2 is to compute normalized arc length, defined as the number of points composing each arc divided by the scale in question. Step 3 is to repeat Steps 1 and 2 for all wavelet scales associated with the wavelet spectrum in question. Repeating the procedure for a finite set of point-wise significance levels allows for the assignment of total arclength to each point in the time-scale plane. The null distribution can be computed using the Monte Carlo approach adopted for the cumulative area-wise test.

The application of the cumulative arc-wise test at the 0.05 significance level to the wavelet power spectra of $X(t)$ and $R(t)$ is shown in Figures 2f and 3f, where point-wise significance levels ranging from 0.02 to 0.18 were used. The discrete spacing between adjacent point-wise significance was set to 0.02 to be consistent with the cumulative area-wise test applied earlier (Figures 2e and 3e). Like with the other statistical tests, the number of spurious point-wise test results for both time series is seen to be dramatically reduced. For $X(t)$, much of the signal detected by the point-wise test remains despite the greater stringency of the cumulative arc-wise test. A comparison of Figures 3c and 3f reveals that the geometric and cumulative arc-wise test detect a similar amount of signal associated with the three sinusoids. As expected, the cumulative arc-wise, like the area-wise test, largely filters out the scale-elongated feature shown in Figure 3b, supporting the idea that both the arc-wise and area-wise tests should be preferred to the other tests in situations in which features associated with periodic signals are sought. In fact, in Appendix A it is shown that the cumulative arc-wise test can detect more signal than the point-wise test for high signal-to-noise ratios.

**2.2.5 Topological Significance Testing**

A common theme among all the statistical procedures discussed so far is that they evaluate the statistical significance of wavelet quantities based on the geometric properties of the patches to which the corresponding points belong. However, as demonstrated by Schulte et al. (2015), topological properties are also important to statistical hypothesis testing. How do topological properties differ from geometric ones? Topological properties of an object are those that remain unchanged no matter how much the object is twisted or bent. Using topological characteristics of point-wise significance regions, additional information about the time series in question can be gained.

The basic principal behind topological significance testing is that topologically equivalent (that is, homeomorphic) objects have identical topological invariants (Ferri, 2017). In the present context, the object of interest is $P_{pw}$ and the topological invariants of concern are the 0- and 1-dimensional Betti numbers denoted by $\beta_0$ and $\beta_1$, respectively. The invariant $\beta_0$ measures the number of path-components or patches composing $P_{pw}$, whereas $\beta_1$ measures the number of 1-dimensional holes in $P_{pw}$. A region in $P_{pw}$ has a hole if there is a closed path in it such that the path cannot be continuously deformed into a point (Figure 4). Equivalently, holes are regions in $\mathbb{H}$ fully surrounded by points in $P_{pw}$. Although counter-intuitive, Schulte et al. (2015) showed that the presence of these holes is related to the statistical significance of the patch to which they correspond. To better understand how these holes are related to statistical significance, consider the reproducing kernel shown in Figure 1b together with the set of points




enclosed by the black contour. As shown in Figure 1b, the closed path, $g$, in the contoured region can be continuously deformed into a point by contracting or shrinking it, with this property holding for any closed path. As such, the set enclosed by the contour has no holes. As patches arise from the reproducing kernel, typical patches arising from isolated fluctuations will tend not to have holes at low point-wise significance levels. However, if two fluctuations are located at nearby frequencies and times, then the resulting patches in the wavelet power spectrum will contain holes (Schulte et al. 2015).

Using Betti numbers, one can assess whether $P_{pw}$ is homeomorphic to $P_{noise} = \{(b, a): \rho_{noise}(b, a) < \alpha_{pw}\}$, where $\rho_{noise}$ denotes the p-value resulting from the application of the point-wise test to a noise spectrum such as the wavelet power spectrum associated with a realization of a red-noise process or the coherence spectrum associated with a pair of red-noise realizations. More specifically, a topological significance test is performed as follows: (1) compute wavelet spectra under some noise model; (2) compute $P_{noise}$ for each of the wavelet spectra; and (3) compute the null distribution of $\beta_0$ and $\beta_1$. The critical values associated with the two-sided topological significance test applied at the $\alpha_{topo}$ significance level are the $100(1-\alpha_{topo}/2)$-th and $100(\alpha_{topo}/2)$-th percentiles of the null distributions. The two-sided test accounts for how the number of patches or holes can be unusually high or low relative to noise. However, as shown by Schulte et al. (2015), a typical patch found in the wavelet power spectrum of red noise will typically have no holes at point-wise significance levels less than 0.2, suggesting that a one-sided topological test may be better than a two-sided test at low point-wise significance levels.

Topological equivalence at a single point-wise significance does not necessarily mean the time series in question is topological indistinguishable from background noise. That is, the result of the topological significance test may be sensitive to the chosen point-wise significance level. To address this concern, it is necessary to adopt the persistent homology approach (Edelsbrunner and Harer, 2008). In this approach, a topological space is distinguished from another topological space homeomorphic to it through an examination of filtered versions of the spaces (Ferri, 2017). In the wavelet analysis context, the space is $\mathbb{H}$ and the filtration is Eq. (13). Computing $\beta_o$ and $\beta_1$ at each step in the filtration results in 0- and 1-dimensional persistent homology profiles (PHP; Qaiser et al., 2016). The more comprehensive topological significance then involves comparing PHPs for a time series in question to that of noise by applying the topological significance test at each point-wise significance level.

The utility of the topological significance testing procedure is demonstrated using the wavelet power spectra of $X(t)$ and $R(t)$. The 1-dimensional PHP corresponding to $R(t)$ shown in Figure 6a indicates that $\beta_1$ is maximum at $\alpha_{pw} = 0.7$ and decreases rapidly until reaching 0 at $\alpha_{pw} = 0.1$. The overall curve is the same for realizations of a red-noise process with any lag-1 auto-correlation coefficient, though the number of holes is greater for larger lag-1 autocorrelation coefficients (not shown). The 1-dimensional PHP for $X(t)$ is like that of $R(t)$, suggesting that these time series are topologically indistinguishable. The application of the topological significance test with $\alpha_{topo} = 0.05$ further supports the topological equivalence of $X(t)$ with red noise because the PHP associated with $X(t)$ falls in the gray-shaded envelope representing the test non-rejection region, where the non-rejection region was obtained by generating 100 PHPs associated with 100 realizations of a red-noise process with lag-1 autocorrelation coefficient





equal to that of $X(t)$. However, $X(t)$ is not noise by construction, and thus the similarity of 1-dimensional PHPs does not exclude the possibility that time series under consideration are indistinguishable from noise.

In contrast to the 1-dimensional PHPs, the 0-dimensional PHP for $X(t)$ and $R(t)$ differ substantially (Figure 6b). The PHP for $X(t)$ is seen to reach a global maximum around $\alpha_{pw} = 0.25$, while the one for $R(t)$ peaks around

$\alpha_{pw} = 0.18$. The application of the topological significance test at the 0.05 level strongly supports the idea that $X(t)$ is distinguishable from noise because $\beta_0$ far exceeds the critical level of the test at $\alpha_{pw} = 0.25$. Thus, there are features that are distinguishable from the background noise. These features are precisely those identified from the area-wise, geometric, cumulative arc-wise, and cumulative area-wise tests.

## 3. Practical Applications to the Indian Monsoon

### 3.1 Data

To understand the temporal behavior and spatial variability of India rainfall, monthly rainfall data for 5 homogenous regions (Parthasarathy et al. 1995a) extracted from Indian Institute of Tropical Meteorology website (http://www.tropmet.res.in) were analyzed. The five divisions are the Peninsula, Northwest, Northeast, Central Northeast and West Central regions. The regions were devised based on continuity of area, contribution to annual

amount, and global/regional circulation parameters (Parthasarathy et al. 1995a; Azad et al., 2010). An all-India (Parthasarathy et al. 1995b) rainfall time series was also examined, as it is commonly reported in the literature. The monthly all-India rainfall time series is constructed by averaging rain gauges at various locations across India (Mooley and Parthasarathy, 1984). These rainfall data span the long time period from 1871 to 2016 and are continuous, making them well-suited for a wavelet analysis. Each rainfall time series was converted into an anomaly time series by

subtracting the 1871-2016 mean for each month from the individual monthly values. After the conversion to an anomaly time series, the time series were standardized by dividing them by their respective 1871-2016 standard deviations. The resulting time series are shown in Figure 7.

The monthly Niño 3.4 index (available at:
https://www.esrl.noaa.gov/psd/gcos_wgsp/Timeseries/Data/nino34.long.data) from 1871 to 2016 was used to

characterize the state of the ENSO system. The Niño 3.4 index is defined as the average SST in the region bounded by 5°N and 5°S and by 170°W and 120°W. The seasonal cycle was removed from this time series in the same way as it was removed from the rainfall time series.

### 3.2 Wavelet Power Spectrum

The wavelet power spectra corresponding to the rainfall time series are shown in Figure 8, where $\alpha_{pw} = 0.05$

in this case. Statistically significant features are seen for all 6 time series. A large swath of point-wise significance is seen around a period of 256 months for the Peninsula, Northeast, and Northwest time series after 1950. For all the time series, most of the point-wise significance patches are seen to be located at lower periods and appear to have varying geometries. For example, the wavelet power spectrum for the Northwest time series contains a scale-elongated significance patch around 1920 extending from a period of 4 months to a period of 64 months. A similar but less





prominent feature is also evident in the wavelet power spectrum of all-India rainfall. For the remaining time series, no such features are readily identifiable from an inspection of the wavelet power spectra.

To account for how spurious results may result from multiple testing, the cumulative area-wise test was applied at the $\alpha_c = 0.05$ level to the wavelet power spectrum shown in Figure 8. For these cases, the cumulative area-wise test was applied using point-wise significance levels ranging from 0.02 to 0.18, with the spacing between adjacent point-wise levels equaling 0.02. Like with the point-wise test, statistically significance wavelet power is located around a period of 256 months for the Northwest and Northeast regions, raising the possibility that these time series contain periodicities at that time scale (Figure 9). For all the time series, most of the statistically significant results are seen at periods less than 64 months. Two of the most prominent features are the scale-elongated significance patches around 1920 in the all-India and Northwest wavelet power spectra. Because the wavelet power associated with these scale-elongated features remain statistically significant after the application of the cumulative area-wise, there is strong evidence that the corresponding fluctuations exceed the red-noise background.

To better differentiate patches arising from singularities from those associated with periodicities, the cumulative arc-wise test was applied to wavelet power spectra shown in Figure 8 at the same point-wise significance levels used for the cumulative area-wise test. As shown in Figure 10, the application of the cumulative arc-wise test at the 5% significance level reveals two time-elongated regions of statistical significance around a period of 256 months in the Peninsula and Northwest wavelet power spectra. However, the wavelet power around a period of 256 months is no longer statistically significant for the Northeast time series, suggesting that the point-wise significant wavelet power at that period is associated with a randomly stable oscillation. The scale-elongated features for the all-India and Northwest power spectra are largely filtered out like the scale-elongated features shown in Figure 3 for $X(t)$. This result suggests that the wavelet power at that time and at those periods is associated with a singularity-like time domain feature rather than a periodicity.

To gain a deeper understanding of the rainfall time series, the 0-th and 1-th dimensional PHPs were computed for each time series. For clarity, we computed anomaly PHPs by subtracting the mean profile associated with noise from the individual profiles at each point-wise significance level. Thus, positive (negative) values mean that the number of patches (or holes) associated with the rainfall time series is greater (less) than that associated on average with noise. Because the lag-1 auto-correlation coefficients associated with each time series are nearly identical, a single test non-rejection region was computed using 100 realizations of a red-noise process with lag-1 auto-correlation coefficient equal to the mean lag-1 auto-correlation coefficient of the 6 rainfall time series. The mean noise profile was subtracted from both the upper and lower bounds of the test non-rejection region after the application of the topological significance test at the 0.05 level.

The $0^{th}$ dimensional anomaly PHPs for each rainfall time series are shown in Figure 11a. Many of the profiles are seen to deviate substantially from the mean noise profiles around $\alpha_{pw} = 0.5$ and $\alpha_{pw} = 0.2$. Around $\alpha_{pw} = 0.5$, the largest deviations are associated with the Northwest, West central, and all-India time series. The profiles for the Northeast and Peninsula time series lay inside the confidence envelope so that those time series could be topologically





indistinguishable from red noise. Around $\alpha_{pw} = 0.15$, the largest deviations are associated with the Northwest and Central Northeast profiles. The large deviations from the mean noise profile for the Northwest time series suggests that the statistically significant features identified by the cumulative area-wise test (Figure 9) are unlikely statistical artifacts. On the other hand, the wavelet power spectrum of the West Central time series contains only 5 small regions

of cumulative area-wise significance despite the large departures from the mean noise profiles at numerous point-wise significance levels. This finding raises the possibility of Type II errors for the cumulative area-wise test or a Type I error for the topological significance test.

Additional insight into the time series is provided by the 1-dimensional anomaly PHPs shown in Figure 11b. In this case, all the rainfall time series appear to be topologically distinguishable from red noise for point-wise

significance levels ranging from 0.7 to 0.65. Like with the 0-th dimensional profiles, the departures are greatest for the Northwest and West Central time series, providing strong evidence that these time series comprise features exceeding background noise. For the West Central time series, the lack of arc-wise significance shown in Figure 10 suggests that the topological significance cannot be attributed to periodicities with high confidence. For the Northwest time series, the cumulative area-wise test results suggest that topological significance may be attributed to the scale-

elongated feature around 1915 and the enhanced wavelet power around a period of 256 months. The PHPs for all the time series converge to 0 below the point-wise significance level 0.3, and the lack of 1-dimensional holes below 0.1 renders difficult the isolation of topologically significant regions in $\mathbb{H}$ using the approach of Schulte et al. (2015) that determines where there is enhanced topological significance based on the location of holes. However, for the Northwest time series, a few holes were identified at point-wise significance below 0.1 around the scale-elongated

patches shown in Figure 9c, supporting the results of the cumulative area-wise test.

**3.3 Wavelet Coherence**

To determine the time scales at which ENSO influences on India rainfall are strongest, wavelet coherence was computed between the Niño 3.4 index and the rainfall time series for each region. The point-wise, cumulative area-wise, and cumulative arc-wise tests were applied to the wavelet coherence spectra at the 5% significance level,

but only the results for the cumulative arc-wise test are shown for brevity. The reason why the results for the cumulative arc-wise test are emphasized is that we are interested in distinguishing non-random coherent oscillations at a scale from randomly stable covarying oscillations.

The results for the wavelet coherence analysis are shown in Figure 12. Statistically significant wavelet coherence was identified for all 6 time series, but coherence between the Niño 3. 4 index and the time series of rainfall

is seen to be most salient for the all-India, Northwest, and West Central regions. The statistically significant coherence is mainly confined to the period band of 16 to 64 months, and time periods of enhanced coherence appear to follow time periods of weak coherence. The results shown in Figure 12 suggests that the 16 to 64-month mode of SST variability across the Niño 3.4 region modulates India rainfall, which is a well-known idea (Torrence and Webster, 1999). It is also noted that the coherence between rainfall and the Niño 3.4 index varies spatially, consistent with

recent work showing how ENSO influences are spatially non-uniform (Roy et al., 2017).



## 4. Summary and Discussion

Wavelet analysis is a tool for extracting time-localized and scale-dependent features from time series. To assess the confidence with which features in wavelet spectra are distinguishable from a noise background, it is necessary to implement statistical significance tests. The traditional approach to statistical significance testing is to

individually test whether a wavelet quantity associated with a point in the time-scale plane exceeds a background noise spectrum. This point-wise approach has been applied in numerous papers since its first application to wavelet analysis by Torrence and Compo (1998).

Despite its wide use, the point-wise test suffers from two pitfalls as noted by Maraun and Kurths (2004) and Maraun et al. (2007) and as shown in Figure 2. The first pitfall is that the test typically produces many false positive

results because of the simultaneous testing of multiple hypotheses. Secondly, spurious results occur in clusters, the clusters reflecting the reproducing kernel of the analyzing wavelet. To remedy the pitfalls of the point-wise test, Maraun et al. (2004) developed an area-wise test to reduce the number of spurious point-wise significance patches based on the area and geometry of the patches. While the method has been shown to be effective at reducing the number of spurious results, the test is computationally inefficient. To address the concern of computational

inefficiency, Schulte et al. (2015) proposed a geometric test that reduces the number of spurious results by assigning a normalized area to each patch. However, both the area-wise and geometric test suffer from a binary decision pitfall in which both an area-wise (or geometric) significance level and a pointwise significance level must be selected. This dual selection inevitably leads to uncertainty regarding the statistical significance of a wavelet quantity.

Schulte (2016) developed a cumulative area-wise test to reduce the sensitivity to the chosen point-wise

significance level. Adopting ideas from persistent homology, the test provides a way to assess the statistical significance of a wavelet quantity located at a point by examining how the area of a nested sequence of patches containing the point changes as the point-wise significance is changed. This test can be viewed as an ensemble technique, where the individual estimates of statistical significance are associated with the p-values of the geometric tests performed at various point-wise significance levels. The ensemble mean is the mean of all such p-values. Much

like how the ensemble mean in weather forecasting filters out unpredictable aspects associated with the individual ensemble members, the cumulative area-wise test filters out the unpredictable aspects associated with a time series in question, the degree to which depends on the chosen cumulative area-wise significance level.

Strictly speaking, the geometric and cumulative area-wise tests (and to a lesser extent the area-wise test) do not assess the confidence with which periodicities are embedded in time series. To make such assessments, the

temporal length of cross-sections of point-wise significance patches needs to be computed and compared to a null distribution of cross-section lengths. This procedure is termed the cumulative arc-wise test and is like the cumulative area-wise test. Two ideal cases showed that the procedure reduces the number of spurious results arising from the point-wise test, while also detecting periodicities embedded in time series. The arc-wise test is also capable of eliminating scale-elongated point-wise significance features induced from singularities in the time domain. Thus,

when testing for the presence of periodicities, the arc-wise test should be preferred to the cumulative area-wise and geometric tests because those tests are unable to distinguish features arising from singularities from those associated





with periodicities. An additional benefit of the cumulative arc-wise test is that it operates in one dimension, rendering its implementation more rapid than that of the cumulative area-wise test.

The application of the statistical tests to the India rainfall time series further highlighted the benefits and disadvantages of the various methods. The application of the point-wise test to the wavelet power spectra of the Indian

rainfall resulted in many statistically significant results. Much of the point-wise significant wavelet power was deemed indistinguishable from background noise after the implementation of the cumulative area-wise and arc-wise tests. However, for some India sub-regions, statistically significant features were identified. For example, the Northwest and all India time series were found to contain features consistent with singularities.

The application of the cumulative arc-wise test to wavelet coherence revealed statistically significant

coherence between rainfall for many of the subregions and the Nino 3.4 index at a period of 16 to 64 months. The most pronounced coherence was found between the all India rainfall and Nino 3.4 index time series, while the weakest coherence was generally found for the Northeast region. These results highlight the spatial variability of the ENSO teleconnection. Periods of high coherence were found to be followed by periods of lower coherence, consistent with prior work showing how the India rainfall-ENSO teleconnection is non-stationary (Torrence and Webster, 1999).

Topological methods were also adopted to further assess whether the rainfall time series are consistent with a red-noise process. An examination of 0-th dimensional PHPs showed that the all-India, Northwest, and West Central rainfall time series are topologically distinguishable from red noise. The 1[th] dimensional PHPs revealed that all rainfall time series are distinguishable from red noise despite the lack of cumulative arc-wise or area-wise significance in the corresponding wavelet power spectra. The discrepancies among the tests suggest a further investigation is required.

An R software package has been written by the author to implement the various tests documented in this paper. The software package can be found at: http://justinschulte.com/wavelets/wavenew.html.



**Appendix A**

A simplified sinusoid experiment was conducted to assess how well the various statistical tests detect a periodic signal embedded in noise. In the experiment, time series of the form

$$X(t) = \sin\frac{2\pi}{T}/\sigma_s + N(t)/m\sigma_n, \tag{A1}$$

were generated. In Equation (A1), $T = 8$ is the period of the sinusoid, $\sigma_s$ is the standard deviation of the sinusoid, $N(t)$ is a realization of a red-noise process, $\sigma_n$ the standard deviation of $N(t)$, and $m$ is a number representing the signal-to-noise ratio. In the experiment, 100 different realizations of $X(t)$ were constructed by generating 100 realizations of $N(t)$. The wavelet power spectrum of each realization of $X(t)$ was computed. The computations were repeated for values of $m$ ranging from 0.01 to 1.0. For each statistical test, the fraction of wavelet power coefficients statistically significant at a period of 8 was calculated for each realization. The mean fraction was then calculated for each value

of $m$, resulting in a curve representing how much signal on average is detected for different signal-to-noise ratios. Values close to 1.0 indicate that the test performed well at detecting the periodic signal. For the experiments, the lag-1 auto-correlation coefficient was set to 0.1, 0.5, and 0.9, but only the results for 0.5 are displayed because the results are only weakly dependent on the lag-1 auto-correlation coefficient. The point-wise, cumulative area-wise, and cumulative arc-wise tests were applied at the 5% level and the geometric and area-wise tests were applied at the 10%

level to 5% point-wise significance regions.

As shown in Table A.1, the amount of signal detected by any of the tests increases as the signal-to-noise increases. Despite how the cumulative area-wise test is more stringent than the point-wise test, the tests detected a comparable amount of signal for low signal-to-noise ratios; for signal-to-noise ratios greater than 0.7, the test

performed better. The cumulative arc-wise test out-performed the cumulative area-wise test for signal-to-noise ratios greater than 0.5. Consistent with theory, both the area-wise and geometric tests performed worse than the point-wise test. The performance of all tests was found to improve if the lag-1 auto-correlation coefficients was increased to 0.9 and was found to worsen if the lag-1 auto-correlation coefficient was set to 0.1.

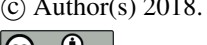



**Data Availability**

Data for India rainfall can be accessed through http://www.tropmet.res.in. The monthly Niño 3.4 index is available at: https://www.esrl.noaa.gov/psd/gcos_wgsp/Timeseries/Data/nino34.long.data.

**Competing Interests**

The authors declare that they have no conflict of interest

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





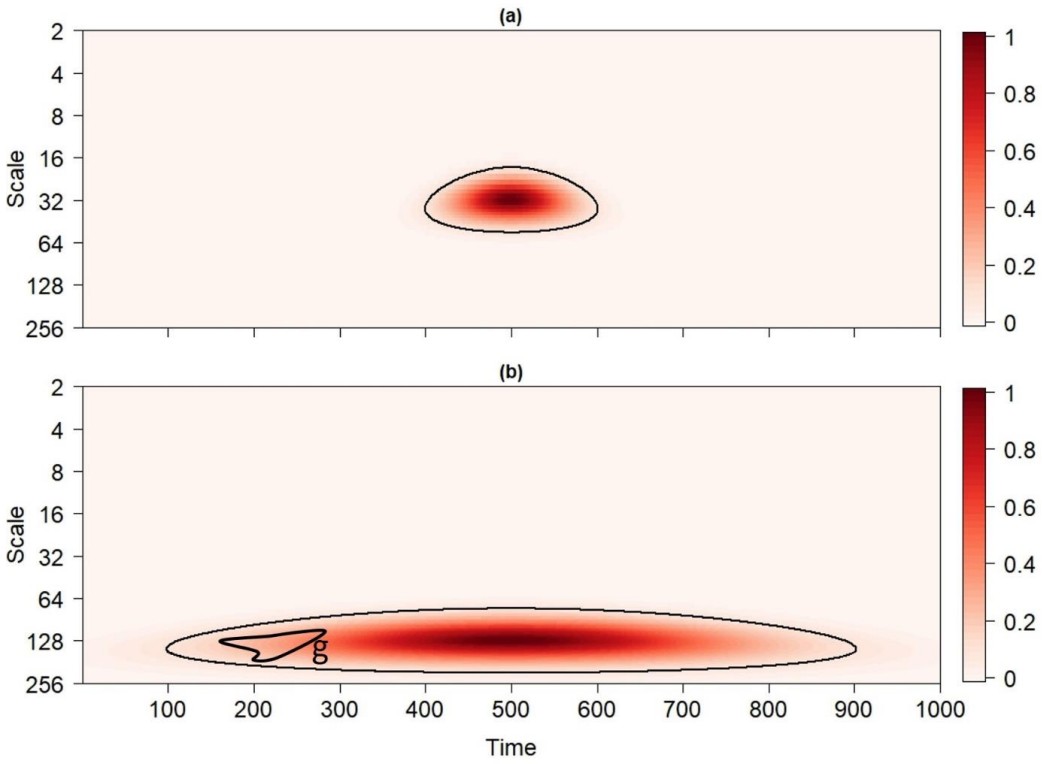

**Figure 1.** The normalized reproducing kernel of the Morlet wavelet dilated and translated to (a) (500, 32) and (b) (500, 128). Contours enclose the regions in the time-scale plane where the normalized reproducing kernel exceeds 0.1. The closed path *g* in Figure 1b can be continuously deformed into a point so that it does not surround a hole in the contoured region.





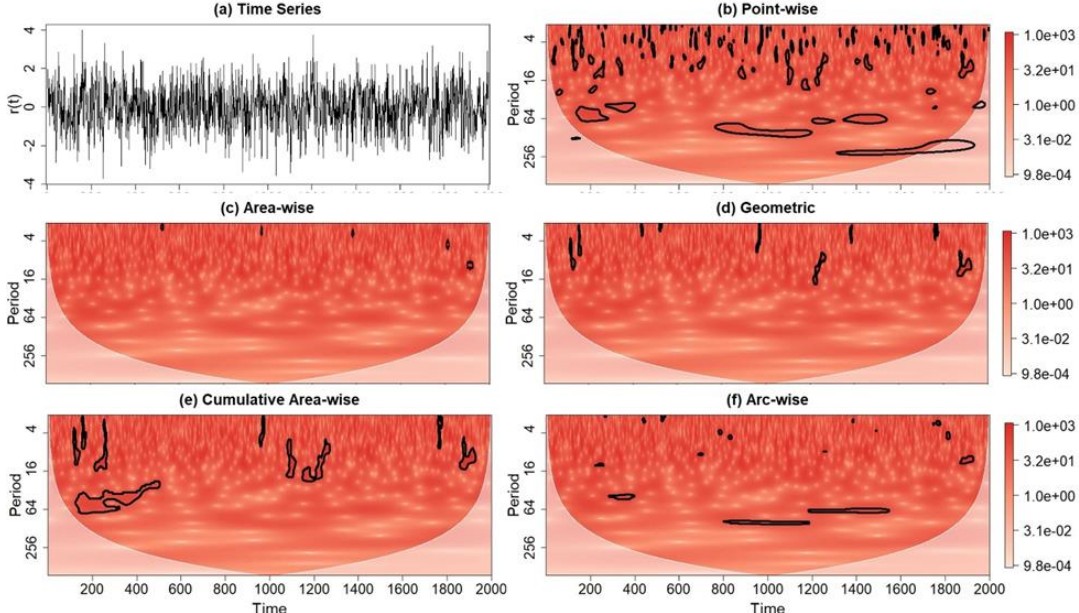

**Figure 2.** (a) A realization of a red-noise process (denoted by $R(t)$) with lag-1 auto-correlation coefficient equal to 0.4 together with its rectified wavelet power spectrum after the application of the (b) point-wise, (c) area-wise, (d) geometric, (e) cumulative area-wise, and (f) cumulative arc-wise tests. The point-wise, cumulative arc-wise, and cumulative area-wise tests were applied at the 5% significance levels. The geometric and area-wise tests were applied at the 0.01 level to 5% point-wise significance regions. Contours enclose regions of statistical significance. Light shaded region represents the cone of influence where edge effects become important.




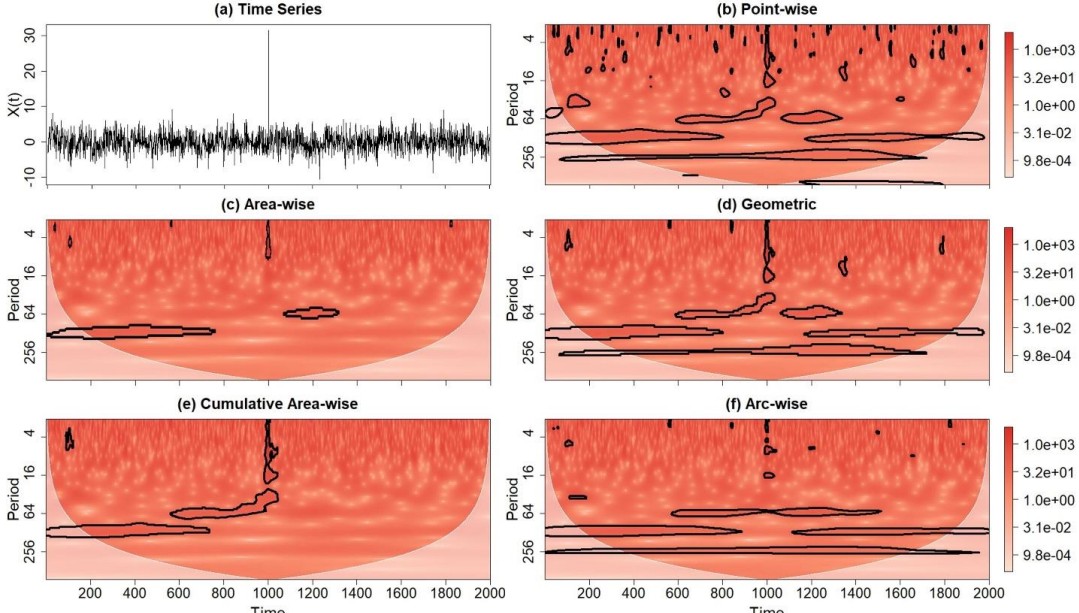

**Figure 3.** Same as Figure 2 except for the ideal time series $X(t)$ representing the linear superposition of three sinusoids with period equal 64, 128, and 256, a singularity at $t = 1000$, and a realization of red-noise process with lag-1 auto-correlation coefficient equal to 0.1. The signal-to-noise ratio, $m$, is equal to 0.4.





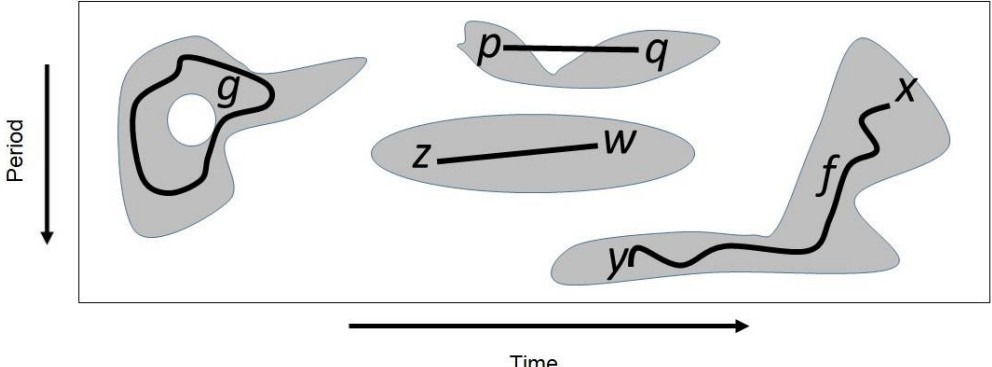

**Figure 4**. Four point-wise significance patches whose union is the set of all point-wise significant values. The closed path $g$ encircles a hole, while the path $f$ connects the points $x$ and $y$ belonging to the same path component or patch. The patch containing the points $z$ and $w$ is convex, contrasting with the patch containing the points $p$ and $q$.

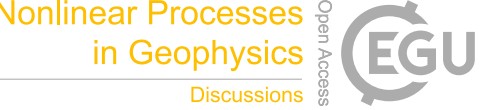

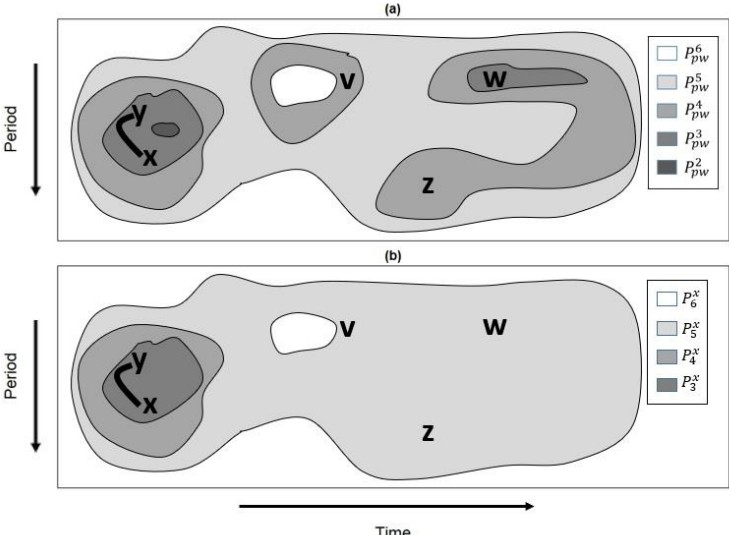

**Figure 5.** (a) An idealized example of a time-scale plane filtration and (b) a geometric pathway corresponding to the point $x$ and the filtration shown in (a). The first two members of the geometric pathway shown in (a) are the empty set and are not shown.





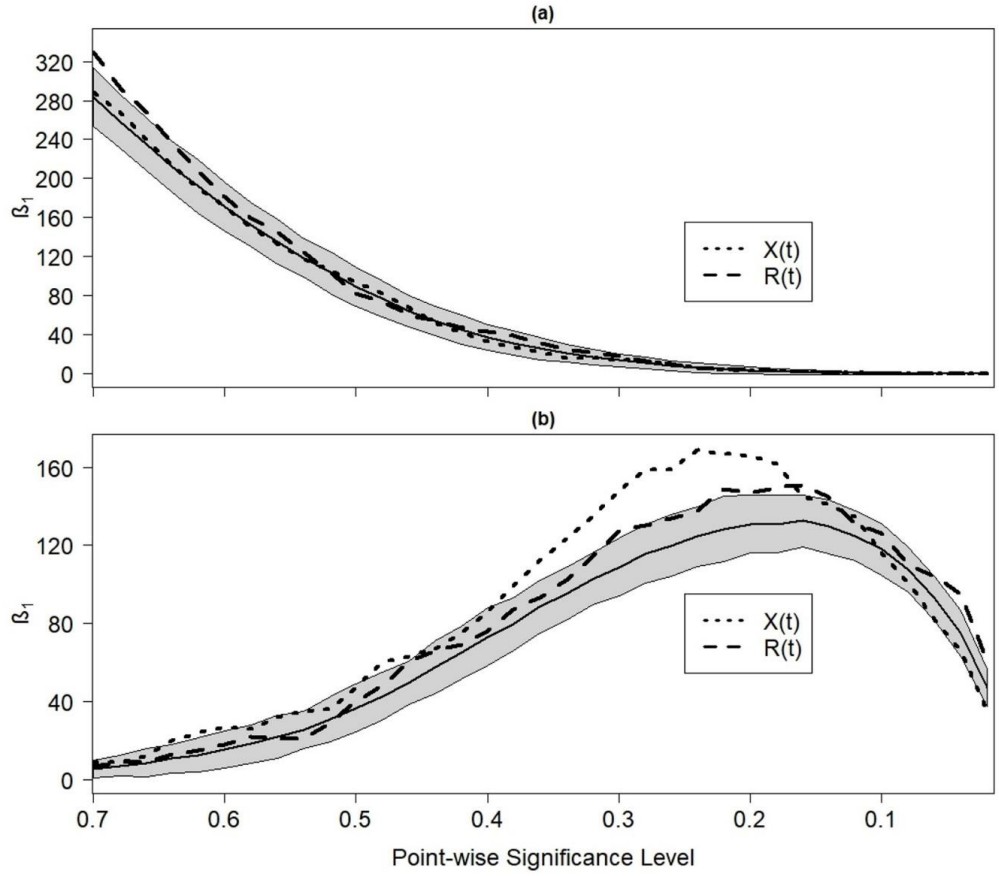

**Figure 6.** (a) The 0-dimesnional and (b) 1- dimensional persistent homology profiles associated with $R(t)$ and $X(t)$. Gray shaded region is the non-rejection region of the topological significance test applied at the 0.05 significance level. The non-rejection region was calculated by generating 100 realizations of a red-noise process with lag-1 auto-correlation equal to 0.1, the lag-1 auto-correlation coefficient corresponding to $X(t)$.





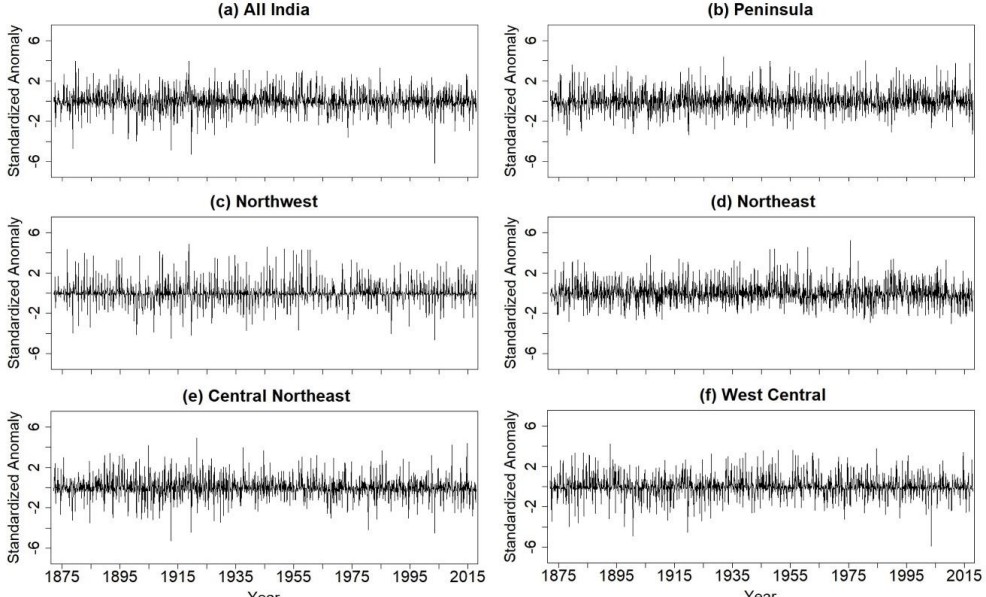

**Figure 7**. Standardized (a) all-India, (b) Peninsula, (c) Northwest, (d) Northeast, (e) Central Northeast, and (f) West Central rainfall time series.




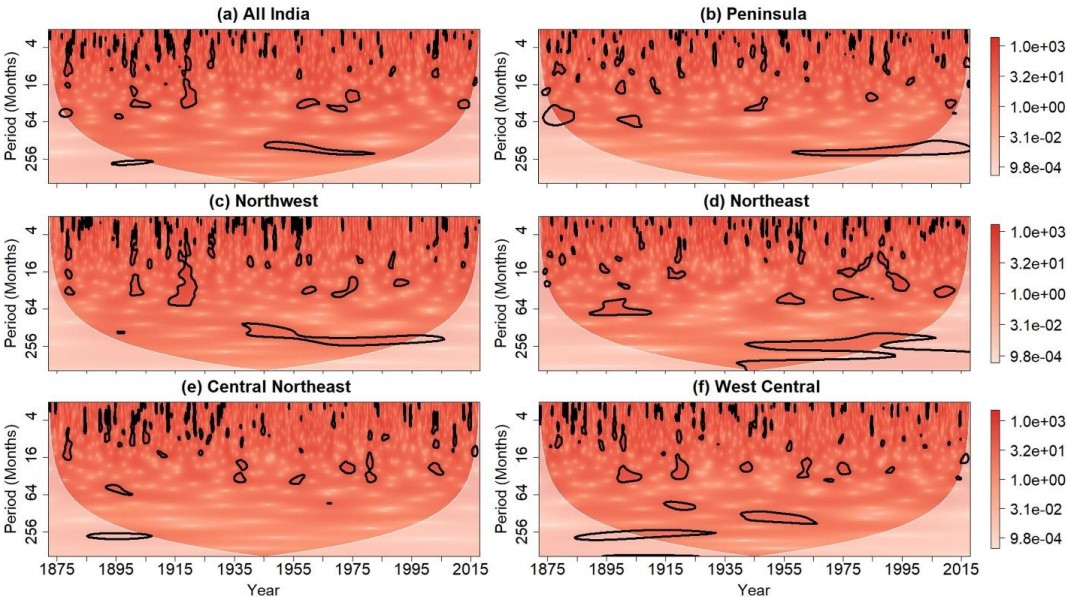

**Figure 8.** Wavelet Power spectra of the standardized (a) all-India, (b) Peninsula, (c) Northwest, (d) Northeast, (e) Central Northeast, and (f) West Central rainfall time series after the application of the point-wise test at the 0.05 significance level. Contours enclose regions of statistical significance and the light shaded region represents the cone of influence where edge effects become important.





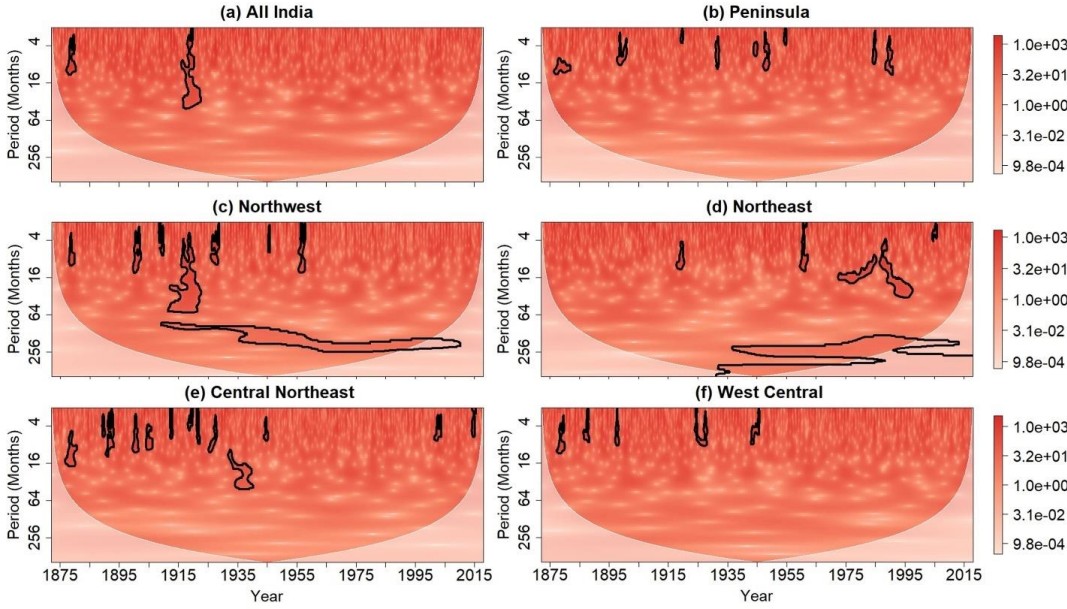

**Figure 9**. Same as Figure 8, but after the application of the cumulative area-wise test at the 0.05 significance level.

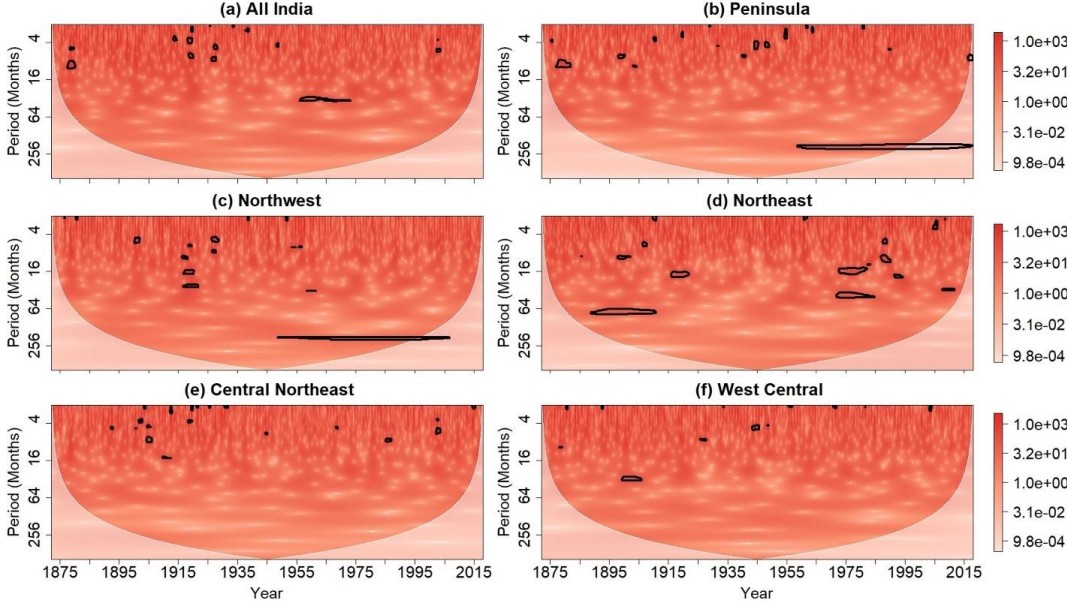

**Figure 10.** Same as Figure 8, but after the application of the cumulative arc-wise test at the 0.05 significance level.

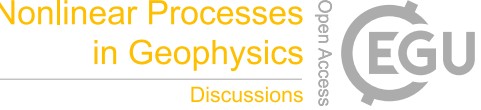



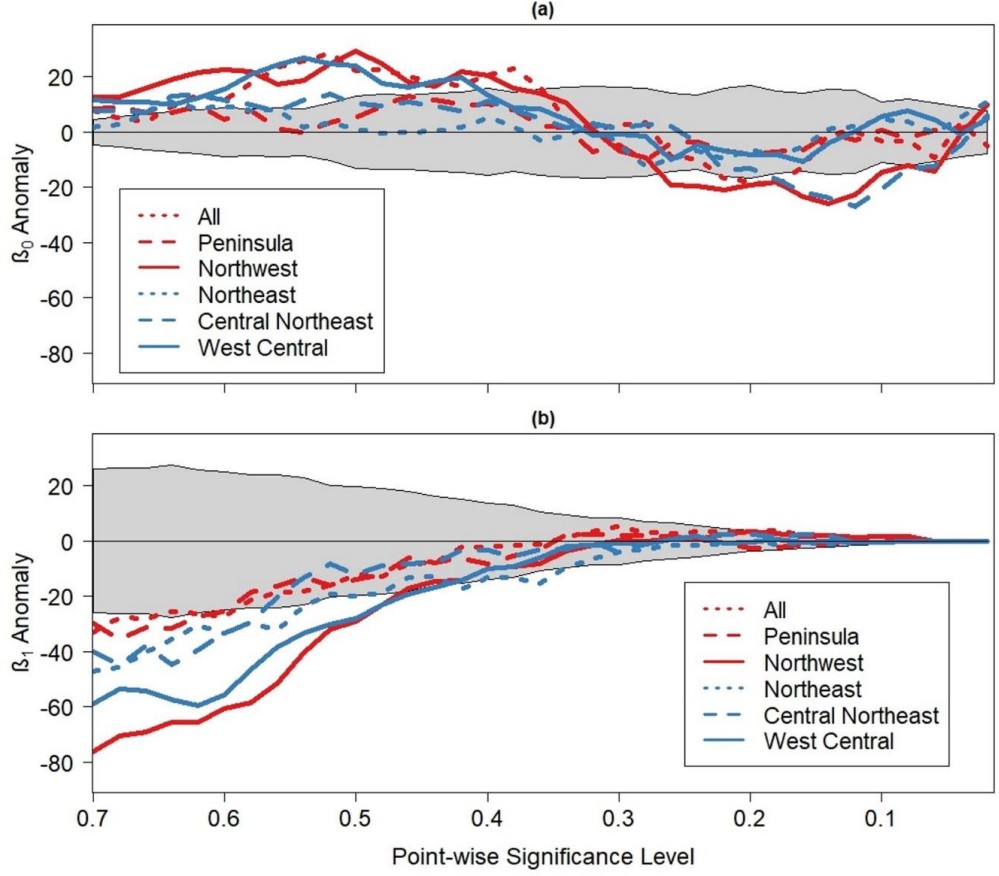

**Figure 11.** (a) The 0-dimesnional and (b) 1- dimensional anomaly persistent homology profiles associated with $R(t)$ and $X(t)$. Gray shaded region is the non-rejection region of the topological significance test applied at the 0.05 significance level. The non-rejection region was calculated by generating 100 realizations of a red-noise process with lag-1 auto-correlation equal to the mean of all 6 lag-1 auto-correlation coefficients associated with the rainfall time series.





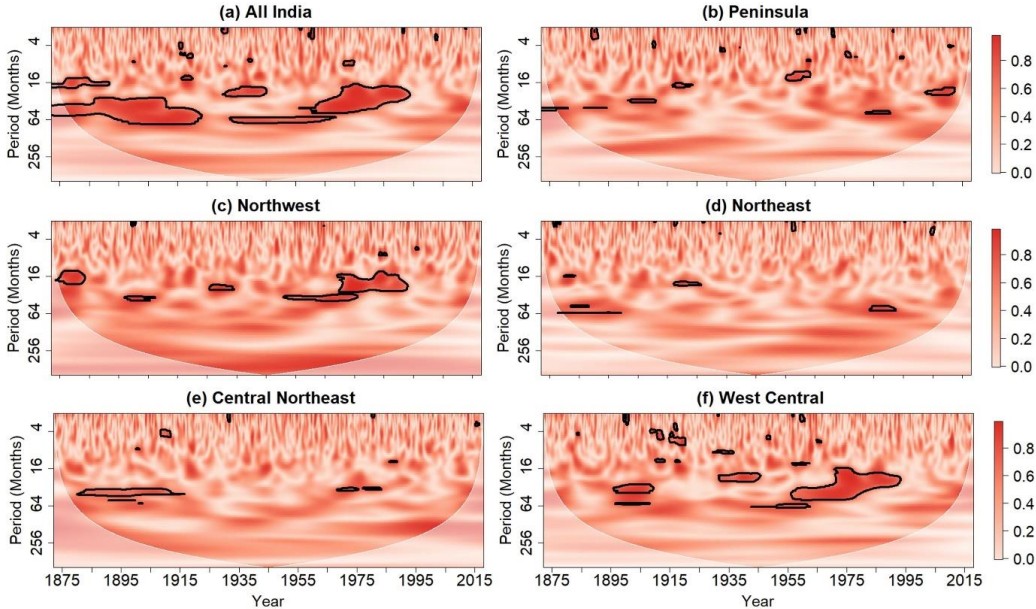

**Figure 12.** Wavelet coherence between Nino 3.4 index and time series of standardized (a) all-India, (b) Peninsula, (c) Northwest, (d) Northeast, (e) Central Northeast, and (f) West Central rainfall. Contours enclose regions where the wavelet coherence is cumulative arc-wise significant at the 0.05 level. For clarity, phase arrows representing the out-of-phase relationship between ENSO and rainfall are not displayed.



**Table A.1.** The fraction of wavelet power coefficients at a period of 8 that are statistically significant in the case of a sinusoid embedded in noise with varying signal-to-noise ratios.

|  | 0.1 | 0.2 | 0.3 | 0.4 | 0.5 | 0.6 | 0.7 | 0.8 | 0.9 | 1.0 |
|---|---|---|---|---|---|---|---|---|---|---|
| **Point-wise** | 0.06 | 0.1 | 0.16 | 0.25 | 0.36 | 0.47 | 0.59 | 0.68 | 0.78 | 0.85 |
| **Area-wise** | 0.01 | 0.03 | 0.05 | 0.12 | 0.22 | 0.33 | 0.44 | 0.69 | 0.69 | 0.79 |
| **Geometric** | 0.03 | 0.04 | 0.07 | 0.16 | 0.24 | 0.36 | 0.49 | 0.65 | 0.74 | 0.84 |
| **Cumulative Area-wise** | 0.04 | 0.07 | 0.12 | 0.19 | 0.33 | 0.47 | 0.62 | 0.76 | 0.85 | 0.92 |
| **Cumulative Arc-wise** | 0.02 | 0.04 | 0.09 | 0.19 | 0.36 | 0.52 | 0.68 | 0.80 | 0.89 | 0.95 |