# Peer review of "Statistical Hypothesis Testing in Wavelet Analysis: Theoretical Developments and Applications to India Rainfall"

_Nonlinear Processes in Geophysics, 2018_

## Referee Comment (RC1) · Anonymous Referee #1 · 2 Feb 2019

General summary The authors have attempted to develop new statistical significance test for wavelet analysis. This is an important contribution as there are many studies involving wavelet analysis and it is important to differentiate between spurious and significant patterns. In addition, a package is developed in R which could be used for testing the proposed statistical method.

I am opinion such a study is of great significance, given the growing application of Wavelet analysis. The theoretical background of conventional point-wise significance testing and the more recent cumulative area-wise method is sound. It provides the reader with an insight into the advantages and drawback of the point-wise method

and the need of cumulative arc-wise testing method. The paper is suggested to be accepted for publication. Below are few comments that would make the proposed analysis more robust and enhance the overall quality of the manuscript:

Major suggestion The authors have attempted to compare the results with the previously published results concerning Indian rainfall. I feel that in order to prove the efficacy of the new method, the author has to apply to many other case studies. Further, it is to be noted that the results (Figure 3) obtained using the arc wise and point wise are comparatively similar and moreover, the latter method is more sensitive to the singularity, the author should provide more evidence for his claim.

Data and reproducibility The authors do not give the complete information on the source and the resolution of the Indian rainfall data. The link of the website (http://www.tropmet.res.in) does not direct to the data page. Authors should provide a complete link of the source of the data, and mention the same in the text to make the work reproducible.

Statement (P13/L11) "To understand the temporal behaviour and spatial variability of India rainfall, monthly rainfall data for 5 homogenous regions (Parthasarathy et al. 1995a) extracted from Indian Institute of Tropical Meteorology website (http://www.tropmet.res.in) were analyzed" does not provide detailed insights about the selection of the data. For example, statement doesn't make it clear whether all stations lying inside the five homogeneous regions were selected, if not then what? Average of all stations lying inside homogeneous regions? Kindly modify the statement appropriately.

I again have a serious concern if mean timeseries of all existing stations within the homogeneous regions has been used. This would result the smoothing of high peaks and might reduce the variability of the rainfall data significantly. Could author comment on the same?

Text and referencing

Although the authors mention that an R package is written, however, the documentation provided in the link (http://justinschulte.com/wavelets/wavenew.html) mentions about the codes in MATLAB only. It would be useful if the authors can provide a direct link of the developed package in R.

Page 1/line 33: wavelet has been applied to broad range of topics... I recently witnessed the drastic use of wavelet in network analysis (for e.g. climate network analysis). Citing and mentioning will link this article to recent study and ultimately I feel it would increase the readability and application of the article.

It would be worth citing few studies based on the same Indian precipitation dataset and wavelet.

---

## Referee Comment (RC2) · Anonymous Referee #2 · 18 Mar 2019

General comments: The paper highlights (and provides the solution to) an important aspect of the application of wavelet analysis which is relevant to the broad field of geophysics and beyond, where an understanding of non-linear and complex processes is required. Correlation among wavelet coefficients is a very well-known issue which has been discussed in the context of forecasting applications using wavelet-based models in many previous studies. However, it is rather rare to see the accommodation of this aspect in wavelet power spectrum and coherence based studies which the author has successfully demonstrated in this paper.

The author provides a novel method of estimating significant periodicities while tacking

the issue of false-positive results employed in differentiating the significant periodicities in the wavelet power spectrum compared to the background noise of the spectrum. The R package for this application would be very useful to the community and I would strongly recommend the community to use, test and validate the proposed approach. The paper is very well written and provides sufficient details and arguments in support of the study. I would recommend accepting this paper pending some very minor corrections which I have listed below:

Technical corrections:

Many sentences are too terse. Especially in the abstract. For eg: "The output of a recently developed cumulative area-wise......." "Statistical hypothesis tests in wavelet analysis are reviewed and developed.: As there may be multiple kinds of analysis pertaining to the application of wavelets, I would suggest specifying what kind of hypothesis test is discussed in that kind of wavelet analysis in this study.

P2 Line 2-3: "To make such comparisons, one must implement statistical tests." Please specify which tests.

P2 Line 11-12: "the first of which is that the test will frequently generate many false positive results because of the simultaneous testing of multiple hypotheses." Please provide relevant references for this point.

P3 Line 3: "a first survey of the theoretical"- I suggest replacing "survey" with "review".

P3 Line 28: what is the significance or the Normalization of the reproducing kernel. are the mathematically different from that shown in Eq 2? Please clarify.

P5 Line 19: Adjective "concrete" is not required.

---

## Author Comment (AC1) · 27 Mar 2019

Summary The author appreciates the detailed comments and suggestions, which have been adopted in the revised manuscript. Changes to the manuscript include the rewriting of sentences to make them less terse, a more detailed discussion about the data sets used in this study, and the inclusion of a supplementary file. More specific changes are outlined below. Reviewer comments are in bold and the author's responses are in plain text. Reviewer 2 General comments: The paper highlights (and provides the solution to) an important aspect of the application of wavelet analysis which is relevant to the broad field of geophysics and beyond, where an

understanding of non-linear and complex processes is required. Correlation among wavelet coefficients is a very well-known issue which has been discussed in the context of forecasting applications using wavelet-based models in many previous studies. However, it is rather rare to see the accommodation of this aspect in wavelet power spectrum and coherence based studies which the author has successfully demonstrated in this paper. The author provides a novel method of estimating significant periodicities while tacking the issue of false-positive results employed in differentiating the significant periodicities in the wavelet power spectrum compared to the background noise of the spectrum. The R package for this application would be very useful to the community and I would strongly recommend the community to use, test and validate the proposed approach. The paper is very well written and provides sufficient details and arguments in support of the study. I would recommend accepting this paper pending some very minor corrections which I have listed below: Technical corrections: Many sentences are too terse. Especially in the abstract. For eg: "The output of a recently developed cumulative area-wise......." "Statistical hypothesis tests in wavelet analysis are reviewed and developed.: As there may be multiple kinds of analysis pertaining to the application of wavelets, I would suggest specifying what kind of hypothesis test is discussed in that kind of wavelet analysis in this study. Many sentences have been lengthened, especially those in the abstract. For example, in the abstract, it is now mentioned that the arc-wise test uses normalized arc length to assess statistical significance. The beginning section of the abstract was also rewritten to provide the reader with some details about the nature of the statistical tests discussed in the study. P2 Line 2-3: "To make such comparisons, one must implement statistical tests." Please specify which tests. The statistical tests are now specified on Page 2, Lines 10 and 11 of the revised manuscript. P2 Line 11-12: "the first of which is that the test will frequently generate many false positive results because of the simultaneous testing of multiple hypotheses." Please provide relevant references for this point. References have been added on page 2 Lines 15 in the revised manuscript P3 Line 3: "a first survey of the theoretical"- I suggest replacing

"survey" with "review". "Survey" has been changed to "review" on Page 3, Line 11 of the revised manuscript. P3 Line 28: what is the significance or the Normalization of the reproducing kernel. are the mathematically different from that shown in Eq 2? Please clarify. Normalization means that the reproducing kernel is divided by its maximum value so that the maximum of the normalized reproducing kernel is equal to unity and located at the point at which the reproducing kernel is centered. Besides division by the maximum value, the mathematical is the same as Eq. 2. P5 Line 19: Adjective "concrete" is not required. The adjective "concrete" has been deleted.

Please also note the supplement to this comment:
https://www.nonlin-processes-geophys-discuss.net/npg-2018-55/npg-2018-55-AC1-supplement.pdf

———————————————————

---

## Author Comment (AC2) · 27 Mar 2019

Summary The author appreciates the detailed comments and suggestions, which have been adopted in the revised manuscript. Changes to the manuscript include the rewriting of sentences to make them less terse, a more detailed discussion about the data sets used in this study, and the inclusion of a supplementary file. More specific changes are outlined below. Reviewer comments are in bold and the author's responses are in plain text. Reviewer 1 General summary The authors have attempted to develop new statistical significance test for wavelet analysis. This is an important contribution as there are many studies involving wavelet analysis and it is important

to differentiate between spurious and significant patterns. In addition, a package is developed in R which could be used for testing the proposed statistical method. I am opinion such a study is of great significance, given the growing application of Wavelet analysis. The theoretical background of conventional point-wise significance testing and the more recent cumulative area-wise method is sound. It provides the reader with an insight into the advantages and drawback of the point-wise method. Below are few comments that would make the proposed analysis more robust and enhance the overall quality of the manuscript: Major suggestion The authors have attempted to compare the results with the previously published results concerning Indian rainfall. I feel that in order to prove the efficacy of the new method, the author has to apply to many other case studies. Further, it is to be noted that the results (Figure 3) obtained using the arc wise and point wise are comparatively similar and moreover, the latter method is more sensitive to the singularity, the author should provide more evidence for his claim. Although the author agrees that the including more cases would better illustrate the efficacy of the statistical tests, the inclusion of more case studies would drastically lengthen the paper. However, many other case studies are included in a new supplementary file for readers to explore. The new case studies provide other scenarios that will further illustrate the efficacy of the methods. For example, a scenario in which a time series is purely noise except for a single large singularity is included. This example has the additional benefit of emphasizing the difference between the point-wise and arc-wise tests. Although the author feels that the arc-wise test results are different from those of the point-wise test, it is agreed that the results are similar enough to make differences hard to discern. As such, the text was reworded to say that the arc-wise test should be preferred to only the cumulative area-wise and geometric tests. Data and reproducibility The authors do not give the complete information on the source and the resolution of the Indian rainfall data. The link of the website (http://www.tropmet.res.in) does not direct to the data page. Authors should provide a complete link of the source of the data, and mention the same in the text to make the work reproducible. A direct link to the webpage from

which the data were obtained is now provided. Statement (P13/L11) "To understand the temporal behaviour and spatial variability of India rainfall, monthly rainfall data for 5 homogenous regions (Parthasarathy et al. 1995a) extracted from Indian Institute of Tropical Meteorology website (http://www.tropmet.res.in) were analyzed" does not provide detailed insights about the selection of the data. For example, statement doesn't make it clear whether all stations lying inside the five homogeneous regions were selected, if not then what? Average of all stations lying inside homogeneous regions? Kindly modify the statement appropriately. I again have a serious concern if mean timeseries of all existing stations within the homogeneous regions has been used. This would result the smoothing of high peaks and might reduce the variability of the rainfall data significantly. Could author comment on the same? The author agrees that more details about the data sets are needed. The details have been incorporated on Page 16. It is now mentioned that the homogenous region rainfall time series are calculated by averaging data corresponding to meteorological sub-divisions after assigning weight to each sub-division based on the area of the sub-divisions. The sub-divisional time series themselves are calculated by averaging the data associated with representative rainfall stations. To the author's understanding, there are about 306 representative stations, the number of such stations differing after 1990. On a similar note, the all-India time series is also created by averaging the sub-divisional data and thus it is based on the approximately 306 representative rainfall stations. Text and referencing Although the authors mention that an R package is written, however, the documentation provided in the link (http://justinschulte.com/wavelets/wavenew.html) mentions about the codes in MATLAB only. It would be useful if the authors can provide a direct link of the developed package in R. The link has been changed to (http://justinschulte.com/wavelets/advbiwavelet.html) Page 1/line 33: wavelet has been applied to broad range of topics. . . I recently witnessed the drastic use of wavelet in network analysis (for e.g. climate network analysis). Citing and mentioning will link this article to recent study and ultimately I feel it would increase the readability and application of the article. The referral to climate network analysis is appreciated.

Three references are included on Page 2 Line 6. It would be worth citing few studies based on the same Indian precipitation dataset and wavelet. Unfortunately, the author could not find any additional studies using the data sets and wavelet analysis after a thorough search.

Please also note the supplement to this comment:
https://www.nonlin-processes-geophys-discuss.net/npg-2018-55/npg-2018-55-AC2-supplement.pdf

---

## Author Response (AR2)

**Reponses to Reviewer 1**

**The suggestions incorporated in the revised version has improved the manuscript from a reader's point of view. Authors considered all my main and minor comments, however, the citations corresponding to climate network and wavelet studies on Indian precipitation dataset are not appropriate.**

The author appreciates the referral to the relevant references, which have been included in the revised manuscript (highlighted in yellow).

**Recent climate network studies based on wavelet**

•       **Paluš, Milan. "Linked by dynamics: Wavelet-based mutual information rate as a connectivity measure and scale-specific networks." Advances in Nonlinear Geosciences. Springer, Cham, 2018. 427-463.**

•       **Agarwal, Ankit, et al. "Wavelet-based multiscale similarity measure for complex networks." The European Physical Journal B 91.11 (2018): 296.**

These references are included on Page 2 and Lines 6 of revised manuscript.

**Studies based on same Indian precipitation dataset and wavelet**

**Well, as mentioned the main source of instrumental meteorological records for India is the India Meteorological Department (IMD). IMD releases dataset in various formats such as gridded dataset covering entire Indian subcontinent, timeseries representing homogeneous regions etc.**

**Few studies based on this dataset are as follows:**

•       **Adarsh, S., and M. Janga Reddy. "Trend analysis of rainfall in four meteorological subdivisions of southern India using nonparametric methods and discrete wavelet transforms." International Journal of Climatology 35.6 (2015): 1107-1124.**

•       **Maheswaran, R., and Rakesh Khosa. "A wavelet-based second-order nonlinear model for forecasting monthly rainfall." Water resources management 28.15 (2014): 5411-5431.**

These references are included on Page 2 and Line 32 of revised manuscript.